# Effect of Protein Surface Hydrophobicity and Surface Amines on Soy Adhesive Strength

**DOI:** 10.3390/polym16020202

**Published:** 2024-01-10

**Authors:** Heikko Kallakas, Nayomi Plaza, Casey Crooks, Derek Turner, Mathew Gargulak, Matthew A. Arvanitis, Charles R. Frihart, Christopher G. Hunt

**Affiliations:** 1Laboratory of Wood Technology, Department of Materials and Environmental Technology, School of Engineering, Tallinn University of Technology, Ehitajate tee 5, 19086 Tallinn, Estonia; 2USDA Forest Service, Forest Products Laboratory, One Gifford Pinchot Drive, Madison, WI 53726, USA; nayomi.plazarodriguez@usda.gov (N.P.); merritt.crooks@usda.gov (C.C.); derek@agchemtech.com (D.T.); matt@agchemtech.com (M.G.); matthew.arvanitis@usda.gov (M.A.A.); charles.r.frihart@usda.gov (C.R.F.)

**Keywords:** soy protein isolate, adhesion, strength, surface hydrophobicity, surface amine group, denaturation

## Abstract

Soy is considered one of the most promising natural materials for manufacturing wood adhesives due to its low cost, high protein content, and ready availability. However, more cost-effective ways of improving its wet shear strength are needed to achieve wider market acceptance. Protein adhesive wet strength depends on the use of (typically expensive) crosslinking additives as well as the processing/denaturation of the protein. It has been commonly stated in the literature that protein denaturation leads to higher bond strength by activating the surface and exposing the reactive groups. Therefore, we investigated how differences in surface reactive groups (surface hydrophobicity and reactive amine groups) brought on with different denaturation treatments relate to bonding performance. Fourteen soy protein isolates (SPIs) with different denaturation histories were investigated. Characterization of the SPIs included surface hydrophobicity, surface amine content, extent of protein hydrolysis, and bond strength (wet and dry, with and without polyamidoamine epichlorohydrin (PAE) crosslinking agent) by ASTM D7998. The molecular weight patterns showed that proteins denatured by extensive hydrolysis had very low bond strengths. Adding the crosslinker, PAE, improved all the shear strength values. We found that the number of water-accessible reactive amine groups on protein surfaces had no impact on the adhesive strength, even with the amine-reactive crosslinker, PAE. Conversely, increased surface hydrophobicity was beneficial to adhesive strength in all cases, though this correlation was only statistically significant for wet strength without PAE. While, in general, denatured proteins are typically thought to form better bonds than native state proteins, this work suggests that it matters how proteins are denatured, and what surfaces become exposed. Denaturation by hydrolysis did not improve bond strength, and extensive hydrolysis seemed highly detrimental. Moreover, exposing hydrophobic surface groups was beneficial, but exposing covalent bond-forming reactive amine groups was not.

## 1. Introduction

There is currently intense industrial interest in biobased adhesives for wood products, as they are considered environmentally friendly alternatives to petroleum-based adhesives. One of the potential biobased resources is soy flour, an agricultural byproduct of soy oil production that is renewable, non-toxic, low-cost, and abundantly available [1,2,3]. Soy-based adhesives have been used in plywood since 1923 [4]. However, by the 1950s, petroleum-based adhesives dominated due to their higher bond strength and water resistance. While soy flour is widely used in North American hardwood plywood [5], low wet bond strength is a major barrier to the wider use of these soy-based adhesives in the wood industry [6,7]. 

Soy protein isolates (SPI) are the most concentrated form of commercially available soy protein (~90+% protein). They are useful for studying protein bonding mechanisms but are too expensive for most wood bonding applications [8,9]. Most commercial SPIs do not represent the proteins in soy flour because SPIs are often hydrothermally denatured [10].

Because of the moisture sensitivity of carbohydrates and good performance available from some purified soy proteins, proteins are commonly assumed to be the soy flour component primarily responsible for water-resistant bonding. Many researchers have reported that wet strengths improve if the soy protein is exposed to denaturing conditions before bond formation [11,12,13,14,15], but others report not seeing this improvement with soy flour. Denaturation is commonly achieved by heat, acid, base, and/or chemical treatments [16]. Denaturation disrupts the thermodynamic balance that keeps proteins in their native state, resulting in a new arrangement of molecular interactions (hydrogen bonds, hydrophobic interactions, etc.). This often results in decreased solubility and increased protein aggregation [17,18].

In the soy adhesive literature, it is commonly stated that denaturing proteins results in stronger bonds in wood because reactive groups are exposed [3,14,19,20]. In our experience, we usually observe cohesive failure in the adhesive rather than adhesion failure. This could potentially be remedied by having stronger protein–protein bonds, for which making reactive groups more available might be useful. The literature often claims that denaturing will unfold the protein molecules into loose and disordered structures, which can increase the surface hydrophobicity and accessibility of amino acid side groups that are available for covalent reactions via the Maillard route or crosslinking agent [3,17,19,20]. These exposed reactive groups, such as amines, are then claimed to increase the shear strength with and without crosslinkers like PAE (polyamidoamine epichlorohydrin) [14,15]. However, since the adhesives are rarely applied to wood under denaturing conditions, the proteins applied to wood are likely non-native but compact, to minimize the contact between the hydrophobic amino acids and the water medium [21]. Because water quickly moves from the adhesive into the wood cell walls after the adhesive application, the adhesive solids content when heat is applied is 50% or more. In a commercial setting, the high solids content limits the protein’s ability to extend during curing. Therefore, we expect the proteins during cure to be fairly compact. The impact of denaturation treatments we apply during this study will be limited to protein refolding/rearranging, resulting in changes to the chemical groups available on the protein surfaces and subsequent aggregation. 

In food science, proteins are often exposed to various denaturing conditions to increase their functionality. In this sense, “functionality” is considered as water holding, fat binding, foaming, etc. [22], which are a function of the physical chemistry and protein arrangement. This differs from the organic chemistry perspective, where “increased functionality” typically means an increase in the number of exposed chemically reactive groups such as amines, thiols, or carboxylic acids. It is reasonable to assume that refolding a protein to increase the number of chemically reactive groups on the surface could increase the density of covalent bonds (and therefore cohesive strength) between neighboring proteins or with crosslinking agents after the bonding process is complete. While it has been shown [15,23] that reactive amine groups such as those on the protein surfaces can react with crosslinkers such as PAE, there is no clear understanding of how the number of reactive amines on protein surfaces impact adhesive performance. Moreover, terms such as “increased functionality” or “reactivity” are often used interchangeably in the literature to explain differences in adhesive performance without specifying what is meant by the term nor providing evidence supporting the claim. 

Protein denaturation can also increase surface hydrophobicity. If hydrophobic domains on neighboring proteins associate during bonding, their association would presumably contribute to the wet strength of the resulting film. Previous studies have shown that increased protein surface hydrophobicity is beneficial for water-resistant adhesive strength when the protein surface has been made hydrophobic by denaturants [13,24]. Alkaline treatment of soy protein has been shown in one study to increase surface hydrophobicity and adhesion strength compared to unmodified soy protein [25].

Here, we addressed the need for evidence to support or disprove the claim that increased functionality leads to better adhesive strength. We investigated the relationship between bond strength and three outcomes of denaturation: reduced protein size via chain cleavage, increased surface amine content, and increased surface hydrophobicity. For this, we used SPI with 14 different denaturation histories. We tested the hypothesis that strong isolates have more reactive groups on their surface and therefore have more ability to react with the crosslinking chemicals and other proteins. The proteins studied included laboratory-isolated SPI with significant native state character, as evidenced by denaturation enthalpy in DSC, as well as thirteen denatured isolates obtained commercially or further reorganized by treatment with ethanol, anoxic dry heating, or wet heating in an autoclave. We then looked for relationships between the number of reactive amine groups on the protein surfaces, surface hydrophobicity, and the extent of hydrolysis on the wet strength, with and without the crosslinking agent, PAE. 

## 2. Materials and Methods

### 2.1. Materials

Five commercial soy protein isolates (SPI), ProFAM 646, 781, 875, 891, and 974, all with protein contents between 85 and 90% with a maximum of 4% fat and 5% ash (see Table 1), were provided by ADM (Decatur, IL, USA). Because commercial SPIs are already denatured, native-state, pilot-plant SPI (PPSPI) was made in the USDA Forest Products Laboratory pilot plant (Madison, WI, USA), as described by Hunt et al. [10]. Briefly, 90 PDI, 200 mesh soy flour Prolia 90-200 (Cargill, Cedar Rapids, IA, USA) was placed in water at 10% solids, pH adjusted to 8.2, and centrifuged to remove insoluble carbohydrates. The supernatant was then acidified to pH 4.5 and again centrifuged to separate the precipitated protein isolate from the soluble carbohydrate supernatant. The precipitated isolate was dispersed as a 15% slurry in reverse-osmosis water, neutralized to pH 7 with NaOH (Sigma Aldrich, St Louis, MO, USA) added dropwise to liquid nitrogen, and freeze-dried. The PPSPI was 85% protein and, when run on SDS-PAGE gels, had a protein molecular weight distribution typical of soy proteins in the literature [26], though some PPSPI aggregates remained in the loading well. A denaturation enthalpy of 11–14 J/g was observed for the PPSPI in DSC (TA Instruments Q20, New Castle, DE, USA), suggesting that about 2/3 of the protein was in its native state [27]. 

Based on the manufacturer’s information, the commercial SPIs were already denatured by jet cooking and possibly other methods to “increase functionality”, i.e., improve food processing characteristics such as emulsification or foaming behavior [22,28]. We increased the number of possible protein configurations by further exposing three protein isolates (781, 974, and PPSPI) to the denaturing conditions of wet autoclave, ethanol soak, and anoxic heat treatment. 

#### 2.1.1. Autoclave Treatment

Dispersions of 15% solids were mixed for one hour, adjusted to pH 7 ± 0.2 using NaOH, if needed, and autoclaved at 121 °C for 15 min. After cooling, the proteins were flash-frozen in liquid nitrogen and freeze-dried in a freeze dryer (Labconco Corporation, Kansas City, MO, USA). No data from the autoclaved 781 is presented because of an error in the sample preparation.

#### 2.1.2. Ethanol Treatment 

SPIs were soaked in 70/30 (*v*/*v*) ethanol/Millipore (10 MΩ or higher resistance, milli-q water systems [29]) water. Five g of protein was placed in a 100 mL beaker with 10 g of ethanol solution and covered for one hour, then uncovered and left in a fume hood to dry. This procedure was chosen because nothing was removed from the protein, no chemical reactions were expected, and aqueous mixtures of low molecular weight alcohols rapidly denature soy globulins [30].

#### 2.1.3. Anoxic Heat Treatment

Approximately 1 g of SPI was weighed into a 50 mL Erlenmeyer flask before being purged with N_2_ for one minute and sealed. The flasks were then placed into an oil bath heated to 120 °C for 15 min and allowed to cool on the bench.

### 2.2. Adhesive Bonding Test

All the soy adhesives contained 15 wt% of SPI in deionized water, stirred by hand with a spatula for 15 min. For half of the samples, 5 wt% of solids (*g*/*g* soy) of commercial crosslinker PAE (polyamidoamine epichlorohydrin, CA 1920) from Solenis LLC (Wilmington, DE, USA) was included in the water. Bond strengths were tested using the Automated Bond Evaluation System (ABES) model 311c (Adhesive Evaluations Systems, Inc., Corvallis, OR, USA) according to ASTM D 7998-19 [31] using sugar maple (*Acer saccharum*) veneers (Columbia Forest Products, Greensboro, NC, USA). The 5 mm × 20 mm overlapped area was hot-pressed for 120 s at 120 °C. After pressing, the samples were stored at 21 °C and 50% relative humidity overnight. Wet shear strength samples were further conditioned by soaking in water for 4 h at room temperature. After conditioning, the samples were tested (tensile shear) in the same ABES system. The ABES method was chosen because it is relatively insensitive to different variables such as solids content, adhesive viscosity, co-solvents to the soy flour, open or closed assembly time, or adhesive spread rate [32]. In addition, we see mostly cohesive failure during wet ABES testing, the same failure mode that dominates in commercial soy-flour-based plywood. The wet strength of the SPIs was tested to understand the critical wet properties, while the dry strength was measured for completeness. Wood failure was not measured because we typically see very little wood failure in wet testing of ABES with soy, and, in dry testing, we see high wood failure at 9 MPa or higher. At least five replicates were tested for every condition.

### 2.3. Molecular Weight Determination

Dry soy protein isolates were dissolved in H_2_O at 0.5 mg/mL for 8 h with occasional heating to 50 °C. The samples were combined with 4× Laemmli sample buffer (1610747, Bio-Rad, Hercules, CA, USA), heated at 100 °C for 10 min, then cooled to 8 °C using a CFX96 thermocycler (Bio-Rad, Hercules, CA, USA). An amount of 10 µg of each sample was loaded onto a 4–20% gradient polyacrylamide gel (4561093, Bio-Rad, Hercules, CA, USA) and run in 1X Tris-glycine-SDS buffer (161732, Bio-Rad, Hercules, CA, USA) at a constant 160 volts. The gels were stained for 2 h with BioSafe Coomassie G-250 (161076, Bio-Rad, Hercules, CA, USA) and destained overnight in water. The molecular weight standards were Precision Plus unstained standards (1610363, Bio-Rad, Hercules, CA, USA).

### 2.4. Surface Amine Quantitation

Fluorescamine dye was prepared by dissolving 100 mg of fluorescamine (Millipore Sigma F9015, St Louis, MO, USA) in 2 mL of dry acetone while inside a glove bag that had been purged with nitrogen. After dissolving, the fluorescamine solution was divided and transferred into airtight sealed vials to minimize exposure to atmospheric moisture.

Sample solutions were prepared by weighing 10–20 mg of the solid protein and mixing it in a 100 mL beaker with sufficient Millipore water to make a solution with a concentration of 0.225 mg/mL. After stirring for 60 min, three 50 µL samples of the solution were transferred into disposable plastic cuvettes, each containing 1950 µL of pH 7.4 phosphate-buffered saline (0.2× standard concentration) and a magnetic stir bar. The autoclaved and ethanol-treated samples required dispersion by sonication; 10.0–12.0 mg of the solid protein was weighed into a 50 mL plastic centrifuge tube, then sufficient Millipore water was added to make a solution with a concentration of 0.45 mg/mL. After being immersed in an ice bath for 10 min, a Qsonica Q125 sonicator (Newtown, CT, USA) with a CL-18 probe was used to disperse the samples. The probe was employed at 45% amplitude for 10 cycles of 10 s each (20 cycles for the ethanol-treated samples). One hour after the start of sonication, 25 µL of the 0.45 mg/mL mixture was dispensed into a disposable cuvette containing 1975 µL of buffer to achieve the same final concentration as the samples made in the primary method.

Fluorescence measurements were taken before and after the addition of fluorescamine for 30 s with active stirring, with an excitation wavelength of 405 nm and an emission wavelength of 480 nm [33]. After the “blank” measurement, 2.5 μL of the fluorescamine solution (~10× excess reagent) was added to each cuvette using a 10 μL glass syringe and immediately agitated. The samples were tested after one hour using the same settings. The final fluorescence intensity was retroactively scaled using actual protein content.

### 2.5. Surface Hydrophobicity (S_0_)

Surface hydrophobicity (S_0_) was determined according to the method of [27] using the fluorescence probe, 1-anilino-8-naphthalenesulfonate (ANS). The ANS dye was prepared by dissolving 50 mg of Mg-ANS (TCI A5353, Tokyo Chemical Industry Co. Ltd., Portland, OR, USA) in 10 mL of 10 mM pH 7.0 phosphate buffer. The stock solutions were 2–3 mg of protein in Millipore water at 0.5 mg/mL. After 1 h, four serial 2:1 dilutions were made (0.25–0.03125 mg/mL).

Fluorescence intensity (FI) was measured with a JY-Horiba Fluorolog Tau-2 (Edison, NJ, USA) spectrophotometer using 365 nm excitation and 484 nm emission, with 30 s of data acquisition while stirring. After taking the blank (protein only) measurements, 20 μL of the ANS solution was added, and the fluorescence was measured after one hour. Protein hydrophobicity was determined from the slope of fluorescence intensity vs. protein concentration calculated by simple linear regression using ordinary least squares methodology. Surface hydrophobicity (S_0_) is a relative measurement, so values obtained on different instruments are not comparable. 

### 2.6. Statistical Analysis

The reported Pearson correlation coefficients (*r*) and associated probabilities (*p*-values) were calculated in MS Excel using data analysis Toolpak/regression to understand the strength and direction of the relationship between variables, such as surface hydrophobicity and shear strength. One-way analysis of variance (ANOVA) and Tukey’s honestly significant difference (HSD) tests were run in R [34] to identify significant differences between material groups. The threshold for statistical significance was 0.05.

## 3. Results and Discussion

### 3.1. Bond Strength

To expand the number of samples with varying strength, surface hydrophobicity, and surface amine content, we exposed some of our isolates to conditions that can change the protein conformation (i.e., denaturing conditions) by exposing PPSPI, ProFam 781, and ProFam 974 to anoxic heat, ethanol, or wet autoclave treatment. The dry and wet shear strengths with and without the crosslinker PAE of the different SPIs as received and after further denaturation treatments are shown (Figure 1 and Figure 2). The ProFam 781 and native state PPSPI stand out for their relatively poor strength under all conditions. All the other SPIs showed moderate to good dry and wet shear strengths without PAE and universal improvement with PAE addition. We commonly observe wood failure in values of 9 MPa or higher, meaning that differences in this range could easily be a result of wood variability. As the wet (water-soaked) strength is typically the most challenging in industrial applications, we focused on the wet bond strengths. For reference, we find that an ABES wet strength of at least 3 MPa is needed for an adhesive to pass the ANSI/HPVA HP-1 [35] standard for plywood [36]. Therefore, most commercial SPIs, without crosslinking agents, delivered wet strength (2.45 MPa or higher) close to minimum commercial acceptance. Adding the crosslinker PAE resulted in an approximately 1.50 MPa wet strength increase for all the SPIs. This agrees with the work of other researchers, where the addition of crosslinker PAE improves the wet strength of soy adhesives [37,38]. However, even with the crosslinker, the ProFam 781 and PPSPI untreated (as produced) still showed wet shear strengths of less than 3 MPa.

The impact of denaturation is also visible in shear strength values. Figure 1 clearly shows that native state PPSPI (furthest point to the right) had lower wet strength than all the other SPIs except ProFam 781. Denaturation of PPSPI resulted in significantly higher wet strength in five out of the six cases shown in Figure 1 (statistical groupings presented in Table A4 and Table A5). All the commercial SPIs were jet-cooked (heavily denatured), and our treatments were very unlikely to return the proteins to their native structure, indicating that almost any of the denatured states are better than native. Interestingly, we can see in Figure 1 and Figure 2 that most of our inexpensive denaturation treatments resulted in PPSPI close to the wet and dry shear strength of jet-cooked ProFam 974 when PAE was included. Despite this apparent convergence of the wet strength of jet-cooked 974 and other denaturation treatments of PPSPI, we still see a statistically significant decrease in the wet strength of five of the six ProFAM 974 samples exposed to further denaturation conditions. This suggests that jet cooking confers some strength improvement relative to other denaturation conditions.

On average, the denaturation treatments increased the native state PPSPI wet shear strength by 1.30 MPa, and by 2.24 MPa when the crosslinker PAE was present. For native state PPSPI, the anoxic heat treatment was notably inferior to the other denaturation treatments when PAE was not present, but after PAE addition, these differences were much smaller. It is clear from Figure 1 and Figure 2 and from previous work [10,39] that all soy protein isolates are not the same with respect to wet shear strength. Also, it is seen that all denaturation treatments do not have the same effect on SPI.

### 3.2. Protein Depolymerization

We looked for evidence of protein depolymerization (hydrolysis) using gel electrophoresis, as depolymerization below a critical point is a well-known method to reduce polymer strength, toughness, and stress crack resistance [40]. Hydrolysis will move intensity from the upper, higher MW portion of a gel (Figure 3) to the lower, low molecular weight portion. Note that the gels were run in a denaturing and reducing environment to minimize protein aggregation. We show the MW patterns of the untreated, as-delivered proteins as they did not change with these denaturation treatments. The ProFam 781 isolate shows a complete loss of high molecular weight bands, indicating that this protein isolate is heavily depolymerized, which we believe explains the low dry and wet strength of 781. All the other SPI have MW profiles that look like the native soy protein [41,42], though some hydrolysis seems to be present in 875, 891, and 974, as evidenced by the low molecular weight bands [42]. Well-defined fractions around the low molecular weight area of 10 kDa have been previously shown also in commercial SPIs [41,42]. The similarity of molecular weights in PPSPI and the other four SPIs with high MW is interesting, because their wet shear strengths are very different (Figure 1 and Figure 2). Depolymerization of proteins tends to increase solubility [43,44]. Therefore, the insoluble protein aggregates in the wells (top of Figure 3) likely contain less hydrolyzed fractions with higher molecular weight. 

Meanwhile, 875, 891, and 974 show evidence of depolymerization, and yet have similar or higher wet strengths than 646 and PPSPI. Therefore, we can say that (1) high levels of depolymerization (20 KDa or smaller fragments) seem to be very bad for wet strength, (2) low levels of depolymerization seem to have a minor impact on wet strength and, (3) parameters other than mild depolymerization have a larger influence on bond strength. 

### 3.3. Surface Amine and Bond Strength

To understand the effect of reactive surface groups (amines) on the bond strength, the amine content values are plotted against shear strength when tested wet (Figure 4) and dry (Figure 5). The first striking feature is that the very highest amine content values come from ProFam 781, the SPI that had been heavily hydrolyzed. In Figure 4 (Left) and Figure 5 (Left), anoxic-heated and ethanol-treated SPI 781 overlap, having identical shear values and amine content. The three treatments lowered the observed surface amine content for the PPSPI and 974 samples, but, in most cases, the differences were not statistically significant (see Appendix A). All the PPSPI and the three treated 974 samples were in the same statistical grouping for amine content. The four samples with the highest amine content were the ProFam 781 and 875, where hydrolysis had created large numbers of terminal amine groups. Despite the often-cited statement “denature to expose reactive groups”, our data suggest that denaturation treatments such as heat or ethanol exposure do not universally increase surface amine content. This is understandable when considering that living plants keep their storage proteins soluble, and solubility is provided by polar groups, such as amines, on the surface [45]. 

Next, we looked at how these exposed reactive amine groups correlate with wet shear strength. The data for all the SPIs combined is skewed by the outlier points from the highly depolymerized, low-strength ProFam 781. However, even after removing 781 from Figure 4 (Left), the remaining data gives a correlation of *r* = 0.07 and *p* = 0.83, suggesting that amine content has no impact on wet strength. 

Even if surface amines are not generally beneficial to wet strength in pure protein formulations, we expected that they would be helpful in the presence of the crosslinking agent PAE, which is expected to be amine-reactive [46,47]. However, after removing 781 (for the reasons discussed above) from the data in Figure 4 (Right), we saw no relationship between wet strength and amine groups on the protein surface, even in the presence of PAE (*r* = 0.07 and *p* = 0.83, the same as without PAE). 

The lack of relationship between surface amine and wet strength might indicate that even the lowest level of amine is already enough to give sufficient reaction with the PAE. Still, primary amine groups in proteins typically have a PKa above 9 [48], indicating that they will have a +1 charge during wet testing of the cured adhesive. We expect primary amines still present in the cured adhesive to attract water and likely weaken the bond during wet testing. Carboxylic acids also react with PAE and contribute to adhesive swelling in water. We expect denaturing conditions that favor exposing polar amine groups will also expose polar carboxyls, meaning that the amine and carboxyl groups are likely to increase in tandem. The carboxyl groups would also be charged at the neutral pH conditions of the wet test, attracting water to the bond line. Therefore, the points relevant to amines should hold for carboxyls as well.

### 3.4. Surface Hydrophobicity and Bond Strength

Another “functionality” potentially exposed on protein surfaces by denaturing conditions is hydrophobicity. We used the standard method [49,50,51] of fluorescence intensity from the hydrophobic fluorescent probe ANS to determine the surface hydrophobic index. The surface hydrophobicity vs. the wet and dry shear strength is shown in Figure 6 and Figure 7. 

As expected, all the denatured proteins had statistically equivalent or higher hydrophobicity than native state PPSPI (see Table A2 for statistical groupings). This is consistent with the general tendency for proteins to precipitate after denaturation. If hydrophobic surfaces are considered reactive groups for making water-resistant hydrophobic associations between neighboring proteins, then this supports the hypothesis that “denaturation exposes reactive groups”. A general increase in hydrophobicity with denaturation could explain why denatured plant proteins typically have better wet bond strength than those in the native state. Hydrophobic surfaces will stick to each other in a water environment, and these hydrophobic associations should contribute to the wet cohesive strength of the adhesive film—if they survive the bonding process [52,53]. Although ovalbumin in egg whites has very different properties than soy proteins, the high wet bond strength of the ovalbumin has been explained by its known ability for a dramatic increase in hydrophobicity (~15× increase in ANS fluorescence) upon heating [54,55]. In our data, there is a slight positive correlation (*r* = 0.39) between SPI surface hydrophobicity and adhesive wet shear strength (Figure 6 (Left)). Removing the anomalous, heavily hydrolyzed ProFam 781 improves the correlation to *r* = 0.69, with a *p* = 0.02, indicating the correlation was unlikely to be observed by chance. We believe we are justified in removing 781 because hydrolysis both destroys wet strength and creates amine groups. When staying within a series generated from the same starting material, the correlations between hydrophobicity and wet strength are even stronger, but the small number of samples reduces the statistical power: four PPSPI treatments: *r* = 0.78, and *p* = 0.22; four ProFam 974 treatments: *r* = 0.85 and *p* = 0.15. No correlation is calculated on 781 because all the variants had 0 MPa wet strength. The dry strength values (Figure 7 (Left)) showed a small positive correlation with hydrophobicity (*r* = 0.39), but the correlation was not statistically significant (*p* = 0.24). 

Because crosslinking agents are very likely to be used in industrial practice, it is important to consider how protein characteristics influence bonding performance in the presence of crosslinking agents such as PAE. The positive effect of surface hydrophobicity on wet strength is still visible when PAE is present (Figure 6 (Right)). After removing ProFam 781, the correlation is 0.59, with *p* = 0.058, slightly short of our threshold for statistical significance, 0.05. This suggests that increasing surface hydrophobicity in soy proteins might be a route to improved wet adhesive performance. This is consistent with the use of the very hydrophobic protein, zein, to make water-resistant films [56]. However, extremely hydrophobic materials such as zein are almost by definition not water-dispersible without cosolvents or extreme conditions, limiting their utility as commercial wood adhesives. 

One might assume that surface amine and hydrophobicity values would be negatively correlated, because amines are charged at neutral pH and therefore hydrophilic, and both occupy the same space, the protein surface. However, there is only a weak correlation between surface amines and hydrophobicity (*r* = −0.18), indicating the two are not mutually exclusive in the range studied. A related question, whether the proteins with less hydrophobic surfaces would benefit more from the crosslinking agent because of potentially more reactive groups on the surface (Table A3), also proved fruitless. The correlation between hydrophobicity and strength improvement with the PAE addition was only *r* = 0.15, *p* = 0.67. 

Though we observed that high levels of surface hydrophobicity and limited protein hydrolysis were associated with higher bond strength, it is likely that other factors, not addressed here, have a major role in determining protein bond performance. While statistically significant, the correlations far from 1 mean that either we have noisy data or are not considering all the drivers of wet strength. One issue is that aggregation can bury significant amounts of a protein’s hydrophobic surfaces [18]. While there is some indication that ANS permeates protein aggregates and adsorbs on interior surfaces [57], it is still possible that some buried hydrophobic domains inside the aggregate would not be observed in the ANS assay, and therefore may help explain the remaining variation in the data. 

## 4. Conclusions

The goal of this study was to investigate the validity of the common sentiment, “denaturation creates reactive groups on the protein surface which result in improved adhesive performance”. We tested the relationship between bond strength and two kinds of reactive groups: surface amine groups and surface hydrophobic domains, as well as the impact of denaturation by protein hydrolysis/depolymerization. We varied the available reactive groups by obtaining a variety of soy protein isolates, many of them denatured in different ways by the manufacturer, and by exposing some of them again to a variety of denaturing conditions. 

We observed a statistically significant positive correlation (*r* = 0.69, *p* = 0.02) between soy protein surface hydrophobicity and wet bond strength when PAE was absent. This correlation was almost unchanged (*r* = 0.59) by the addition of 5% (dry *w/w* on soy) of the crosslinking agent, PAE, but did not reach the threshold for statistical significance (*p* = 0.058). 

We did not observe any increase in strength due to protein hydrolysis, and the heavily hydrolyzed SPI with no proteins above 20 kDa had universally very poor strength, even when the crosslinking agent, PAE, was added. Therefore, from this data set, we cannot attribute any strength benefit to denaturation by protein hydrolysis but can say extensive hydrolysis appears to be bad for bond strength.

We observed no significant correlation between water-accessible reactive amine groups on protein surfaces and adhesive strength, even in the presence of the amine-reactive crosslinker, PAE. The lack of correlation likely indicates that all the protein conformations had sufficient reactive groups available for PAE reaction. This data supports the hypothesis that denaturation improves soy protein bond performance by exposing reactive groups—if hydrophobic surfaces are considered reactive groups. Effects of denaturation other than those studied here also likely contribute to bond strength development.

## Figures and Tables

**Figure 1 polymers-16-00202-f001:**
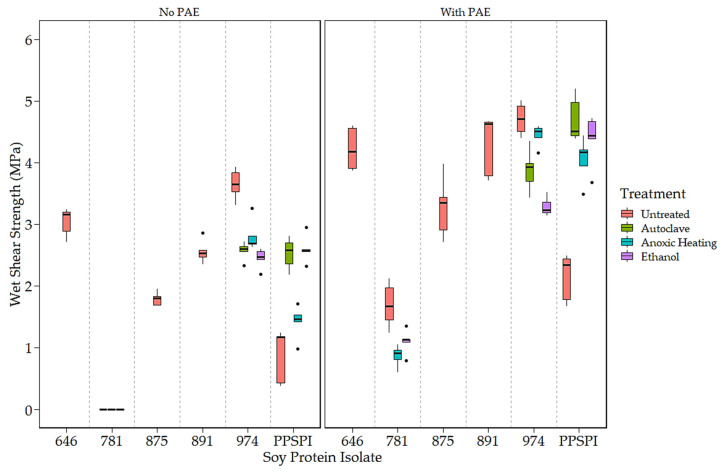
Wet shear strength of SPI in 14 different states of denaturation, (**Left**) without and (**Right**) with the crosslinker, PAE (All 781 without PAE delaminated before testing. Data points identified as outliers are shown as black dots).

**Figure 2 polymers-16-00202-f002:**
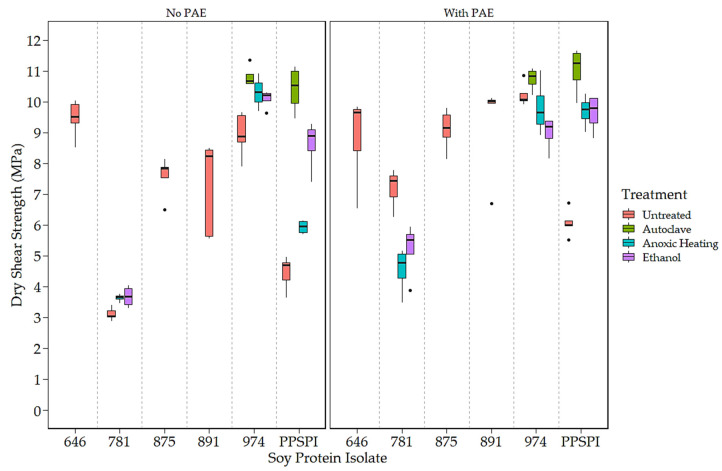
Dry shear strength of SPI in 14 different states of denaturation, (**Left**) without and (**Right**) with the crosslinker, PAE.

**Figure 3 polymers-16-00202-f003:**
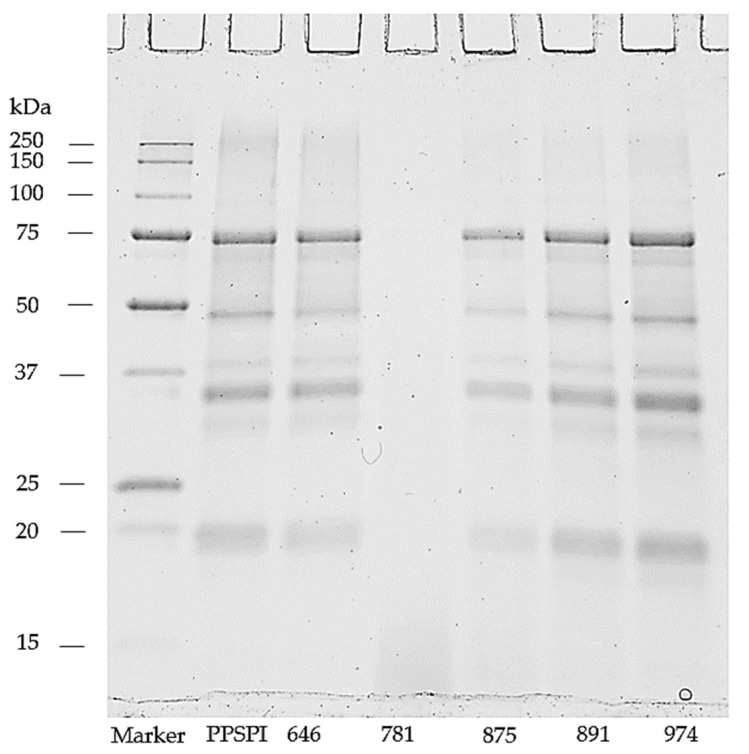
Soy protein isolates resolved on reducing SDS-PAGE gel.

**Figure 4 polymers-16-00202-f004:**
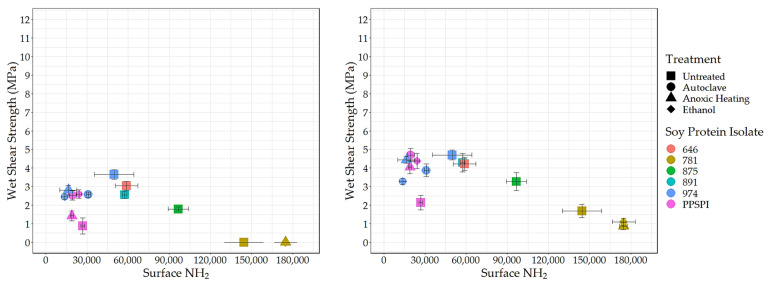
Surface amine content (relative value) vs. wet shear strength. **Left**: no PAE, **Right**: with PAE.

**Figure 5 polymers-16-00202-f005:**
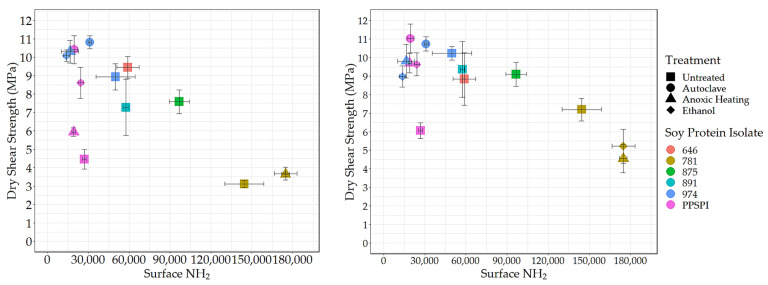
Surface amine content (relative value) vs. dry shear strength. **Left**: no PAE, **Right**: with PAE.

**Figure 6 polymers-16-00202-f006:**
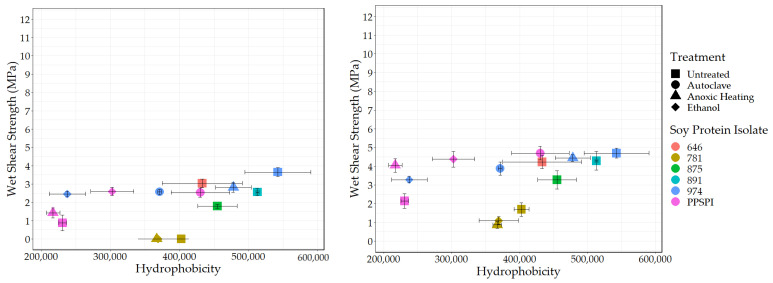
Surface hydrophobicity vs. wet shear strength. **Left**: no PAE, **Right**: with PAE.

**Figure 7 polymers-16-00202-f007:**
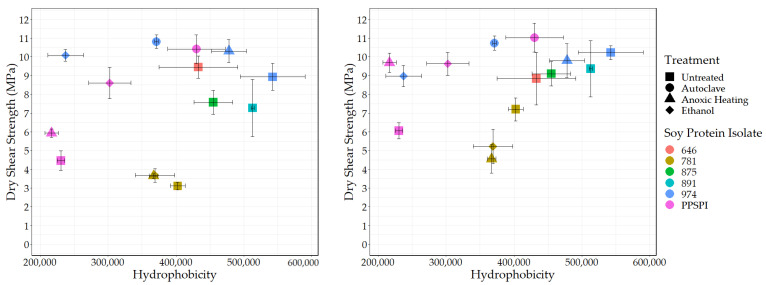
Surface hydrophobicity vs. dry shear strength. **Left**: no PAE, **Right**: with PAE.

**Table 1 polymers-16-00202-t001:** Composition and treatments of SPIs.

Soy	Treatment	Protein (%)	Ash (%)
PPSPI	Untreated	84.75	4.07
PPSPI	Autoclave	84.75	4.07
PPSPI	Anoxic Heating	84.75	4.07
PPSPI	Ethanol	84.75	4.07
974	Untreated	88.55	4.17
974	Autoclave	88.55	4.17
974	Anoxic Heating	88.55	4.17
974	Ethanol	88.55	4.17
891	Untreated	88.67	4.94
875	Untreated	87.08	4.96
781	Untreated	88.67	4.84
781	Ethanol	88.67	4.84
781	Anoxic Heating	88.67	4.84
646	Untreated	90.13	3.63

## Data Availability

The data presented in this study are available on request from the corresponding author. The data is not publicly available due to the confidentiality of the running project.

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
