# Peer review of "Effect of Protein Surface Hydrophobicity and Surface Amines on Soy Adhesive Strength"

_polymers, 2024, doi:10.3390/polym16020202_

Round 1

Reviewer 1 Report

Comments and Suggestions for Authors

The authors do a commendable job relating surface hydrophobicity, surface amine content, extent of protein hydrolysis, and bond strength both wet and dry of the various SPI’s and conditions tested.  

There is a lot of good research being conducted around soy adhesives but there are still may questions about the mechanism of how these adhesives work. The authors  presented data that supports their analysis goal of determining if surface hydrophobicity has an effect on wet strength of soy adhesives. They also efficiently showed that water accessible reactive amine groups on protein surfaces did not contribute significantly to adhesive strength, something that is stated often in the literature.

The one area of the paper that could be improved are the figures which are complicated and require significant concentration to get their full meaning. Perhaps simplifying the figures in some way would be helpful.

Author Response

Please find the authors responses to reviewer comments in Word file.

Reviewer 2 Report

Comments and Suggestions for Authors

This article presents a study on effect of protein surface hydrophobicity and surface amines on soy adhesive strength. Fourteen soy protein isolates with different denaturation histories were investigated. There are several questions not mentioned or clearly clarified by the authors, so the manuscript should be revised before publication.

1) How you designed your experiment? Explain the methodology.

2) Why were the six original soy protein isolates not treated identically in the experiment?

3) Line138: pH 7.0 +-.2 is written correctly?

4) Line143: It should be explained about "Millipore water".

5) Line277 and 278: We commonly observe wood failure in values of 9 MPa or higher, meaning that differences in this range could easily be a result wood variability. should be placed in the Results and Discussion.

6) Line322: Anoxic heated and ethanol treated SPI 781 overlap, having identical dry shear values and amine content. should be placed in the Results and Discussion.

7) Conclusion: give significant number data.

8) The spacing of numbers and units in manuscripts needs to be standardized.

9) The language should be checked carefully throughout the manuscript, there were some mistakes and obscure sentences.

10) please improve the quality of figures.

11) References need to be modified in MDPI format.

12) References: [8] and [20] contain obvious errors.

Comments on the Quality of English Language

Minor editing of English language required

Author Response

(The authors gave the same response as above.)

Reviewer 3 Report

Comments and Suggestions for Authors

Dear editor, dear authors,

 The present work is one of the most complete study on protein adhesives that I have ever seen. The authors consider several products and also different denaturations.

The work is absolutely well presented, clear and interesting and shed clear light on a controversial topic which is very up-to-date in the field of bio-based wood adhesives.

The only suggestion I might give is to consider the addition of a table in the materials paragraph (2.1) were: i) the difference between ProFAM 646, 781, 875, 891 and 974 are highlighted and ii) the denaturation on PPSPI summarized.

Author Response

(The authors gave the same response as above.)
